# An Aqueous Exfoliation of WO_3_ as a Route for Counterions Fabrication—Improved Photocatalytic and Capacitive Properties of Polyaniline/WO_3_Composite

**DOI:** 10.3390/ma13245781

**Published:** 2020-12-17

**Authors:** Mariusz Szkoda, Zuzanna Zarach, Konrad Trzciński, Andrzej P. Nowak

**Affiliations:** Faculty of Chemistry, Department of Chemistry and Technology of Functional Materials, Gdansk University of Technology, Narutowicza 11/12, 80-233 Gdansk, Poland; zuziaz696@gmail.com (Z.Z.); trzcinskikonrad@gmail.com (K.T.); andnowak@pg.edu.pl (A.P.N.)

**Keywords:** electrodeposition, PANI/WO_3_ composite, supercapacitor, photocatalytic properties

## Abstract

In this paper, we demonstrate a novel, electrochemical route of polyaniline/tungsten oxide (PANI)/WO_3_) film preparation. Polyaniline composite film was electrodeposited on the FTO (fluorine-doped tin oxide) substrate from the aqueous electrolyte that contained aniline (monomer) and exfoliated WO_3_ as a source of counter ions. The chemical nature of WO_3_ incorporated in the polyaniline matrix was investigated using X-ray photoelectron spectroscopy. SEM (scanning electron microscopy) showed the impact of WO_3_ presence on the morphology of polyaniline film. PANI/WO_3_ film was tested as an electrode material in an acidic electrolyte. Performed measurements showed the electroactivity of both components and enhanced electrochemical stability of PANI/WO_3_ in comparison with PANI/Cl. Thus, PANI/WO_3_ electrodes were utilized to construct the symmetric supercapacitors. The impact of capacitive and diffusion-controlled processes on the mechanism of electrical energy storage was quantitatively determined. Devices exhibited high electrochemical capacity of 135 mF cm^−2^ (180 F g^−1^) and satisfactory retention rate of 70% after 10,000 cycles. The electrochemical energy storage device exhibited 1075.6 W kg^−1^ of power density and 12.25 Wh kg^−1^ of energy density. We also investigated the photocatalytic performance of the deposited film. Photodegradation efficiencies of methylene blue and methyl orange using PANI/WO_3_ and PANI/Cl were compared. The mechanism of dye degradation using WO_3_-containing films was investigated in the presence of scavengers. Significantly higher efficiency of photodecomposition of dyes was achieved for composite films (84% and 86%) in comparison with PANI/Cl (32% and 39%) for methylene blue and methyl orange, respectively.

## 1. Introduction

Organic–inorganic composites have been widely researched, as they are able to combine benefits from the both components and simultaneously overcome their individual drawbacks. It makes them interesting for many possible applications [1,2,3,4,5]. As a result of the synergetic effects, enhancing of electrical, electrochemical, photocatalytic or mechanical properties can be observed, and therefore their utilization in various fields like supercapacitors, sensors, photocatalysis and rechargeable batteries is achievable [6,7,8,9,10,11]. In the case of such hybrids used as electrode materials, conductive polymers-based systems deserve close attention.

Conductive polymers are of great interest as potential electrode materials due to their high accessible surface area (high porosity), relatively high conductivity and high capacitance even at very high charge/discharge rates [12,13]. Polyaniline (PANI) is one of the most extensively used conducting polymer for supercapacitors. However, PANI is characterized by a relatively narrow potential window of electrochemical activity and low stability during multiple charge–discharge cycles [14,15]. An extended potential range of electroactivity and an improvement of specific capacitance can be achieved by the fabrication of organic–inorganic composites with metal oxides or composites with carbon nanomaterials [16,17,18]. Commonly, nanoscopic engineering or obtaining nanocomposites based on polyaniline (PANI) and another active materials remarkably improves PANI-based supercapacitors’ performance and PANI-based photocatalysts [16,19,20,21]. Nevertheless, many factors affect the working properties of the polyaniline-based devices: its oxidation and protonation state, and the doping level and the dopant’s character [22]. Recently, the topic of the counterions has gained much more attention and one can find a considerable amount of work on the study of its impact. Neoh et al. [23] reported that the extent of the anion penetration of the PANI’s film relied strongly on the anion size, showing the protonation of the ClO_4_^−^ and TSA (toluene sulphonic acid) anions to a significant depth of the base film, while the SSA (sulphosalicylic acid) and DBSA (dodecylbenzenesulfonicacid) anions were limited to the surface areas only. Abd-Elwahed and Holze [24] observed that PANI properties were highly affected by the size of anions, but yet it was not the only factor responsible for regulating the process of dedoping and redoping. Other research stated that PANI has a “size memory effect”, indicating that during the electropolymerization of aniline in a protonic acid, PANI may retain the size of the doping anions being introduced, and can only be redoped by the same or smaller anions [24,25]. However, the impact that counter ions may have on the electrochemical properties of PANI has remained vague. One of the recent studies [26] showed that it was possible to improve the charge–discharge cycling stability of polypyrrole and polyaniline by depositing a carbonaceous shell on them. Moreover, the results indicated that the electrodes featured particularly high capacitance retention, though the mechanism was not entirely understood.

Carbonaceous material can be embedded into the conductive polymer matrix as a counterion, as it was shown for graphene oxide (GO) [27]. GO flakes suspended in water carry a negative surface charge that allows them to act as counter ions during PEDOT (poly(3,4-ethylenedioxythiophene)) and polypyrroleelectropolymerization [28]. On the other hand, we elaborated the method of WO_3_ aqueous exfoliation that leads to the formation of a negatively charged suspension of WO_3−x_, abundant with the surface groups [29]. Based on the idea of using GO as a counterion source, here we performed aniline electropolymerization in the presence of WO_3−x_, without the addition of other electrolytes. The developed method allowed the film of PANI/WO_3_ to be electrodeposited directly on the conductive substrate from an aqueous electrolyte. Since both materials, PANI and WO_3−x_, exhibit electrochemical activity and electric storage ability [30,31,32], PANI/WO_3_ composite was investigated here as an electrode material for supercapacitors. Furthermore, the combination of two photoactive semiconductors (p-type PANI and n-type WO_3_) was expected to be a way of improving the photocatalytic properties of the composite. Thus, the electrodeposited PANI/WO_3_ layer was investigated as a photoactive material for the decomposition of organic dyes (methylene blue and methyl orange).

## 2. Materials and Methods

### 2.1. Preparation of PANI/WO_3_ Film

All materials used in this work were purchased from Sigma Aldrich (Saint Louis, MI, USA). They were of analytical grade and used without further purification.

The procedure leading to formation of a transparent solution containing hydrated and amorphous WO_3_nanoflakes was described in our previous report [29]. To sum it up briefly, it is based on a one-stage, environmentally friendly and facile water-based strategy, which is described in the literature as exfoliation.

Polyaniline/WO_3_ film was electrodeposited onto FTO (fluorine-doped tin oxide) substrates using a chronoamperometry method. The 50 mL of aqueous solution used for the deposition of a composite film contained 0.2 mL of exfoliated tungsten oxide (WO_3_; 200 μL) and aniline (ANI) monomer (200 mM). For comparison, PANI was also deposited from an electrolyte containing 3 M HCl and ANI monomer. The reference and counter electrodes used for the electrodeposition were Ag/AgCl/3M KCl and platinum mesh, respectively. The potential of aniline polymerization was determined on the basis of linear sweep voltammograms, presented in Figure 1a. The onset potential of ANI oxidation was shifted (over 0.3 V) towards more cathodic potential for an electrolyte containing exfoliated WO_3_. Since the onset potential of aniline oxidation was lowered and the addition of acid is not necessary, the range of possible substrates for electroplating of polyaniline increased. These results show that the exfoliated WO_3_ facilitated the polymerization of ANI compared with Cl^−^ as the counterion. A similar effect has been already observed for EDOT electrooxidation. Sakmeche et al. [33] observed a lowered polymerization potential for EDOT in the presence of sodium dodecyl sulfate, which was explained by strong electrostatic interactions between monomer radicals and dodecyl sulfate anions.

The mass loading of the investigated electrode materials was obtained using analytical balance RADWAG XA 82/220.4Y.A PLUS (Radwag, Radom, Poland). The values derived from the independent measurements were equal to 0.75 ± 0.12 mg cm^−2^ and 0.88 ± 0.15 mg cm^−2^ for PANI/Cl and PANI/WO_3_, respectively.

An exemplary chronoamperometry curve recorded during PANI/WO_3_ electropolymerization on a FTO is presented in Figure 1b. Charge consumed for electrodeposition of polyaniline (for both type of counterions) was equal to 1.0 C cm^−2^, unless otherwise stated. Thus, the preparation procedure of PANI/WO_3_ organic/inorganic composite consists of (i) WO_3_ aqueous exfoliation and (ii) PANI/WO_3_electrodeposition. The diagram depicting the strategy of a hybrid material preparation is presented in Figure 1c.

### 2.2. Physicochemical Characterization Techniques

The morphology of the electrode materials was investigated using scanning electron microscopy (FEI QUANTA FEG 250) (Quanta 3D FEG, Fei Company), equipped with secondary electron detector in the high vacuum mode (pressure 10^−4^ Pa). 

The XPS analysis was performed using Argus Omicron NanoTechnology X-ray photoelectron spectrometer (Scienta Omicron GmbH, Limburger, Germany) with an Mg-Kα source. The measurements were conducted under ultra-high vacuum at room temperature, under a pressure below 1.1 × 10^−8^ mbar. The data was analyzed with CASA XPS software package using Shirley background subtraction and a least-squares Gaussian–Lorentzian curve fitting algorithm. The obtained spectra were calibrated to give a binding energy of 285.80 eV for C1s.

### 2.3. Electrochemical Studies

The electrode materials were tested using cyclic voltammetry (CV) and galvanostatic charge–discharge tests (CP) using the Bio-Logic VSP potentiostat–galvanostat (Biologic, Seyssinet-Pariset, France). In a three-electrode configuration, the Ag/AgCl/3.0 M KCl and Pt mesh served as the reference and the counter electrode, respectively. The electrochemical measurements were carried out under argon atmosphere in 1 M H_2_SO_4_ aqueous solution, which was purged with argon for 45 min before every measurement in order to remove oxygen. A symmetric supercapacitor was constructed by combining two FTO/PANI/WO_3_ electrodes and placing a fiberglass separator soaked in 1 M H_2_SO_4_ aqueous electrolyte between them. In the next step, the heat sealable foil was welded on three sides using a plastic foil welder and eventually it was sealed using a vacuum packing machine (Henkelman, Hertogenbosch, Netherlands). A sticky copper tape (3M) was used as an electric contact. The supercapacitor cells were tested using multiple galvanostatic charge/discharge cycles (i_c_ = i_a_ = 3 mA/cm^2^).

### 2.4. Measurements of Photocatalytic Activity

To evaluate the photocatalytic activity of the PANI/WO_3_ and PANI/Cl layers, the degradation rate of methylene blue (MB) and methyl orange (MO) dyes in an aqueous phase was monitored. A FTO glass covered by an active material with the surface area of 1 cm^2^ was placed at the bottom of the reactor, filled with 25 mL of an aqueous solution of MB and MO (initial concentration (C_0_) =10 μM). A xenon lamp, equipped with an AM 1.5 filter with a light intensity of 100 mW cm^−2^ was used as the light source. By holding the mixture for 30 min in the dark, the adsorption–desorption equilibrium was attained. Thereafter, 0.75 cm^−3^ of the dye solution was withdrawn at a regular time interval. The UV–VIS absorption spectra of the solution were analyzed using the UV–Vis spectrophotometer (Biosens, Warsaw, Poland) model UV5100 (Metash) in 1 cm (the length of the light path equals to 1 cm) quartz cuvettes to evaluate the efficiency of the photodegradation process.

## 3. Results and Discussion

### 3.1. Morphology and Chemical Structure

The morphology of the FTO/PANI/Cl and FTO/PANI/WO_3_ was investigated through SEM analysis. The electrodeposition of PANI/Cl onto FTO leads to the formation of an electrode material in the form of rather smooth nanofibers with a diameter of about 150 nm (Figure 2a). SEM micrographs suggest high microporosity of the deposited film. As it can be concluded on the basis of micrographs presented in Figure 2d,e, the presence of WO_3_ had a significant impact on the film’s morphology. Polyaniline fibers cannot be simply distinguished (see Figure 2d). A higher magnification allows one to observe the nanostructured nature of the obtained material, see Figure 2e. The WO_3_ part is well embedded and dispersed uniformly on the polymer matrix. The cross-sections presented in Figure 2b,c confirm that the materials layers, whose thickness is almost 9 µm thick for PANI/WO_3_ and 15 µm for PANI/Cl, are homogenous and spongy throughout its volume. 

The EDX spectrum (Figure 2f) consists of peaks related to the presence of carbon, nitrogen, tungsten and oxygen atoms in the composite film. The weight content of elements in the material was found to be as follows: C (62.76 wt %), N (23.15 wt %), O (4.67 wt %), W (6.15 wt %) and Sn (3.27 wt %). The presence of tin is characteristic for FTO substrate. The results give insight into the quantitative ratio between polyaniline and WO_3_.

The high-resolution XPS spectra recorded for the PANI/WO_3_ film in the energy range characteristic for W4f, C1sand N1s are presented in Figure 3. Figure 3a shows the wide-scan XPS spectra of the as-prepared material. The spectrum consists of the bands characteristic for N and C from polyaniline and W from WO_3−x_. No other elements were detected, confirming that only exfoliated WO_3_ was incorporated as a counterion into the polymer structure during the electrodeposition. The C1s spectrum (Figure 3b) consists of two broad peaks, one at 284.8 eV and another peak at 286.3 eV [34,35]. The peak at 284.6 eV shows the presence of a quinoid structure and benzenoid rings. The N1s spectrum shows one broad asymmetric peak at 399.8 eV, which signifies the presence of more than one type of nitrogen, see Figure 3c. After deconvolution two peaks can be distinguished. The peak at 399.6 eV can be attributed to the benzenoid structure, while the peak appearing at 401.8 eV is probably due to the presence of quaternary ammonium salt structure [35]. The XPS spectrum of the W4f region is presented in Figure 3d. The doublet at 38.1 and 36.1 eV was recorded and ascribed to the binding energies of the W4f 5/2 and W4f 7/2 orbital electrons of W^6+^, respectively. According to our previous report about exfoliated WO_3_, exfoliation procedure leads to the formation of WO_3−x_ enriched in surface groups, where tungsten occurs in mixed oxidation states (V and VI) [29]. Here, the reduced form of W was not detected, which is probably the effect of the preparation procedure. Anodic polarization of the substrate during electrodeposition results in the oxidation of the monomer, and the counterion. Since no other elements that could originate from impurities and act as a counterions were detected in the XPS wide-range spectrum (see Figure 3a) and aniline was successively polymerized during anodic polarization, it can be assumed that positive charges of oxidized polymer chains were neutralized by exfoliated WO_3_. 

### 3.2. Electrochemical Properties

#### 3.2.1. Three-Electrode Configuration

The cyclic voltammetry measurements of FTO/PANI/Cl and FTO/PANI/WO_3_ electrodes were carried out to investigate and compare their electrochemical properties (Figure 4a). The obtained materials exhibited characteristic peaks of redox couples of polyaniline [36]. Generally, the current density recorded for a PANI/WO_3_ electrode material was significantly larger than that of the PANI/Cl (both electroactive films were obtained using the same charge consumed for the electrodeposition). This effect may be observed due to the much more developed surface of WO_3_-containing film (see Figure 2d) and additional electroactivity that comes from an inorganic element of the composite. It may provide a larger number of active centers on the surface of a film. This could lead to a better contact with the electrolyte and thus to an enhanced charge transfer process.

The multiple charge/discharge cycles were performed in order to test the obtained electrodes as possible electrodes materials for energy storage devices, namely supercapacitors. The areal capacitance (*C_A_*) and volumetric capacitance (*C_V_*) were calculated using the discharge time from galvanostatic charge–discharge curves using the Equation (1):(1)$$C_{A\;or\;V}=\frac{I\cdot dt}{dV\cdot A\;or\;V}$$
where *I* is the applied discharge current, *t* is the discharge time, *V* is the range of electrode potential, *A* is the active area in cm^2^ and *V* is the volume of the electrode material. Areal and volumetric capacitance in the function of the cycle number is presented at Figure 4b,c, respectively. The largest decrease in capacity for both electrodes was observed in the first 1000 cycles. It was observed that the capacity value stabilizes in subsequent cycles. The areal capacitance (*C_A_*) was 215 mF cm^−2^ (*C_V_*= 302 F cm^−3^) and 84 mF cm^−2^ (*C_V_* = 64 F cm^−3^) at a current density of 2 mA cm^−2^ for FTO/PANI/WO_3_ and FTO/PANI/Cl, respectively (after 2500 cycles). The capacity retention is one of the most important factors in determining the supercapacitor performance. The values of 74% and 49% of *C_A_* and *C_V_* were reached after 4000 cycles for FTO/PANI/WO_3_ and FTO/PANI/Cl, respectively. In order to investigate the effect of a capacitance drop, the SEM image of the electrode surface after 1000 charge/discharge cycles was performed. As it is shown in Appendix A, the PANI/WO_3_ electrode material retains its morphology. Thus, it is unlikely that the decrease of capacitance is related to the leach of components from the electrode material. According to the previous paper about electrochemical performance of exfoliated WO_3_, the capacitance retention of inorganic part should be high [29]. Additionally, when comparing CV curves recorded for PANI/WO_3_ electrode before and after 1000 cycles (see Appendix A), the disappearance of the peaks related to the oxidation/reduction of polyaniline can be observed. Thus, it is very likely that the drop of capacitance is mainly related to the decrease of pseudofaradaic activity originating from PANI.

The decrease in performance of the PANI/WO_3_ composite was attributed to the degradation of PANI, as was evidenced by Wei et al. [37]. According to the obtained results, it may be concluded that the formation of an organic–inorganic composite had a positive influence on PANI electrochemical performance. The presence of WO_3_ not only increased the capacity of the electrode material, but also positively affected its cycling stability.

In order to investigate the mechanism of the energy storage using PANI-based electrodes, the cyclic voltammetry measurements at different scan rates were performed. The results of the CV measurements are shown in Figure 5a (for FTO/PANI/Cl) and Figure 5d (for FTO/PANI/WO_3_) for a potential window from 0 to 0.8 V with scan rates of 10, 20, 50, 75, 100 and 200 mV s^−1^. The shape of all CV curves at the different scan rates exhibited redox couple activity, possibly indicating the presence of pseudocapacitive characteristics of polyaniline electrodes. To determine the mechanism responsible for the charge transfer process, plots j = *f*(v) (Figure 5b,e) and j = *f*(v^1/2^) (Figure 5c,f) were performed at the defined potential value (E = 0.6 V) for both electrode materials. For PANI/Cl electrode material, obtained fitting results showed that both mechanisms wereequally responsible for charge accumulation process, while in the case of PANI/WO_3_ contribution of charge storage by pseudocapacitance is much greater in comparison with the diffusion controlled mechanism. Despite the fact that the basic analysis was performed, more detailed investigation was necessary to identify the mechanisms responsible for the charge storage.

In the next step, it was to distinguish whether the electrode process is capacitive (surface mechanism) or diffusion dependent in its nature. It is known that the total current response of an electrode material is a sum of these two mechanisms [38]:(2)$$i\left(V\right)=\;k_{1}\cdot v+k_{2}\cdot v^{1/2}$$

where $k_{1}\cdot v$ and $k_{2}\cdot v^{1/2}$ are attributed to the current contributions from capacitive effects and diffusion-controlled process, respectively. Thus, knowing *k*_1_ and *k*_2_ parameters, one is able to quantify, at a defined potential, the fraction of the current response regarding each of the contributions. For an analytical purpose, Equation (2) can be transformed to Equation (3):(3)$$i\left(V\right)/v^{1/2}=\;k_{1}\cdot v^{1/2}+k_{2}$$

However, in a real system a mixed process is the most common and such an approach gives information that no simple mechanism occurs. It is due to the fact that charge storage is not only occurring on the electrode’s surface but also in the bulk material. Figure 6 shows the results of this analysis for three different scan rates: 10, 50 and 100 mV s^−1^ for PANI/WO_3_ and PANI/Cl electrode materials. The shaded regions are attributed to the capacitive currents, whereas the white regions are attributed to diffusion-controlled processes. In the case of PANI/Cl electrode material the fraction of the capacitive contribution is similar to the fraction of a diffusion-controlled process, especially for 50 mV s^−1^. It is very likely that conducting polymers with small mobile counterions store energy through fast and reversible redox reactions between an electrolyte and electrode material. This phenomenon causes that both capacitive and diffusion-controlled mechanisms seem to be rate-independent during the electrochemical process at higher scan rates.

For PANI/WO_3_ the charge stored at *v* = 10 mV s^−1^wasalmost equally divided between the surface mechanism and the diffusion-controlled mechanism, while for PANI/Cl one may see the dominant effect originating from diffusion-controlled process. At higher sweep rates the total charge was mostly stored utilizing a capacitive mechanism for PANI/WO_3_ and almost equally mixed process was observed for PANI/Cl. It was evidenced that the presence of WO_3_ affects the type of storage mechanism of the electrode material. For PANI/WO_3_ the contribution of the surface mechanism increased with the sweep rate increase, confirming that for higher sweep rates the kinetics were not diffusion-limited but were limited by surface processes. This was in agreement with the results shown in Figure 5e. Moreover, looking at Figure 6d, there was no shaded region at the current maxima. It evidences that for redox couple activities of polyaniline, the charge storage mechanism was controlled by the diffusion process for low sweep rates values.

The results of the analysis show that the implementation of WO_3_ in the polymer matrix increased the overall capacitance of the electrode material mainly using the surface mechanism, which is not so depended on the scan rate. Thus, PANI/WO_3_ electrode material that exhibits higher capacitance and stability than PANI/Cl can be a promising material for supercapacitors.

#### 3.2.2. Two-Electrode Configuration

A symmetric supercapacitor was fabricated using two FTO/PANI/WO_3_ electrodes. The electrochemical characterization of the supercapacitor was performed using galvanostatic charge–discharge cycles. The device was fabricated using FTO/PANI/WO_3_ electrode material obtained using different charge consumed for electrodeposition. The charge/discharge experiment was performed using 1.0 mA charge/discharge current and 0.7 V voltage range. The highest value of specific capacity was achieved for the highest charge value at electropolymerization, due to the thicker film of PANI/WO_3_ (see Figure 7a). In the case of the film obtained using 0.05 C cm^−2^, after 1000 cycles, the capacity retention was equal to 83% (from 2.75 to 2.3 mF cm^−2^), for 0.5 C cm^−2^, the capacity retention was at the level of 62% (from 87.5 to 54 mF cm^−2^), whereas for 1 C cm^−2^ was equal to 77% (from 175.5 to 136 mF cm^−2^). Despite the fact that the stability of the electrode material was lower for 1.0 C cm^−2^ in comparison with 0.05 C cm^−2^, the value of the specific capacity was higher for the former. Therefore, the film electrodeposited using 1.0 C cm^−2^ was chosen for further investigations (multiple charge–discharge cycles). Before that, the most appropriate current density was chosen based on the results presented at Figure 7b. Additionally, charge/discharge profiles for a PANI/Cl symmetric supercapacitor are presented at Figure 7c. The curves have a typical capacitors triangular shape. The results indicate that higher capacitance values were obtained for a PANI/WO_3_ supercapacitor, as the discharge time was almost twice higher at the same current density value. Moreover, a PANI/Cl supercapacitor was not able to work at as high current rates as the PANI/WO_3_ due to the immediate discharge process, which is a limiting factor for the practical application.

Multiple charge–discharge cycles were performed in order to test the stability of the PANI/WO_3_-based supercapacitor. As it is shown in Figure 8a, after the capacitance drop after about 1000 cycles, very good stability up to 10,000 cycles was obtained. The total capacitance retention between the 1st and the 10,000th cycle was equal to 69%. It is noteworthy that the decrease of capacitance was the highest for the first 1000 cycles and then the capacitance stabilized, reaching 91% retention between the 1000th and the 10,000th cycle. The areal capacitance of PANI/WO_3_ symmetric supercapacitor was 135 mF cm^−2^ at a current density of 3 mA cm^−2^ and the gravimetric capacitance was 160 F g^−1^ at a current density of about 1.0 A g^−1^ (after 1000 cycles). For a comparison, the gravimetric, volumetric and areal capacitance obtained for a PANI/Cl symmetric supercapacitor (at a current density of 1 mA cm^−2^, for which the discharge time was similar as for the PANI/WO_3_ supercapacitor) are presented in Appendix A at Appendix A.

Additionally, the energy density (ED) and an average power density (PD) were calculated from the discharge curve according to the following Equations (4) and (5) [39]:(4)$$E_{D}=\frac{C_{s}{\left(\cdot U\right)}^{2}}{2}\frac{1000}{3600}\left(\frac{Wh}{kg}\right)$$
(5)$$P_{D}=\frac{E_{D}}{t}\left(\frac{W}{kg}\right)$$

where *C_s_* is the specific capacitance, ΔU is the voltage range and *t* is the time of discharge. Hence, the energy density value of 12.25 Wh kg^−1^ and the power density 1075.6 W kg^−1^ were obtained for a PANI/WO_3_-based supercapacitor. The Ragone plot shows the power and energy densities, compared with the reference values (see Figure 8b) [40,41,42,43,44,45,46,47].

### 3.3. Photocatalytic Properties of the PANI/WO_3_ Composite

In addition to electrochemical measurements, the photocatalytic activity of the electrodeposited PANI/WO_3_ composite was investigated and compared with PANI/Cl film. The photocatalytic degradation of methylene blue (MB) and methyl orange (MO) using PANI/WO_3_ and PANI/Cl electrode materials as photocatalysts was performed and the results are presented at Figure 9a,b. In order to investigate the effect of irradiation on the catalyst/dye interaction, two sets of experiments were carried out. For the first one, based on the direct photodegradation of the dye, no significant changes of dyes concentration were observed. The decolorization efficiency of the MB achieved 9% after 2 h illumination, while the decolorization efficiency of MO was found to be less than 3%. The results showed that the photolysis of the dyes was rather low and MO was more stable under the illumination. Then, the photo decolorization of dyes was performed in the presence of PANI/WO_3_, and PANI/Cl and the photocatalytic effect of both material was confirmed. However, the PANI/WO_3_ exhibited significantly higher photoactivity towards organic compounds decomposition in comparison with PANI/Cl. The results are summarized in diagram in Figure 9c. One may see that the WO_3_ played an important role in increasing the photocatalytic activity of PANI, as the decolorization efficiency increased more than 2 times for both dyes in comparison with PANI/Cl. The enhancement of the photoactivity may result from the formation of the junction between the organic and inorganic component. Since the polyaniline exhibits p-type conductivity and WO_3_ is an n-type metal oxide semiconductor, it may be expected that the rate of e^−^/h^+^recombination on the interface will be significantly inhibited. Therefore, photoexcited charges can be more efficiently consumed for the organic dyes degradation. In order to investigate the role of WO_3_ more deeply, the dye decomposition was investigated in the presence of selective scavengers. 

The results of the photocatalytic degradation of MB by FTO/PANI/WO_3_ in the presence of different scavengers are presented in Figure 10a. The obtained results indicated that the addition of ammonium oxalate (AO), whichwas used as a hole (h^+^) scavenger, significantly inhibited the photodegradation efficiency of MB decomposition. Thus, it is very likely that the degradation of MB was strictly related to direct photooxidation of dye adsorbed on the photocatalyst via holes from the valence band. PANI/Cl was not able to decompose MB directly using holes due to the location of the valence band [48]. It can be concluded, that the presence of n-type WO_3_ facilitated the degradation of the organic dye via direct oxidation using holes from the valence band of WO_3_. Moreover, after the addition of benzoquinone (BQ), which is commonly used as a superoxide radical scavenger, the MB photodecomposition efficiency was at the similar level as in the case of the experiment without the addition of scavengers. Thus, superoxide radicals did not participate in the photodegradation of the organic dye. Therefore, the main chemical individuals responsible for dye photodegradation were photoexcited holes from the WO_3_ valence band. Since the MB decomposition efficiency decreased in the presence of t-butyl alcohol (TBA), the hydroxyl radicals participated in the dye decomposition, as well. Thus, the presence of WO_3_ enhanced the oxidative properties of the inorganic–organic composite.

To evaluate the photocatalytic stability of the catalyst, PANI/WO_3_ was used in four photocatalytic experiments (on the example of MB degradation). The results in Figure 10b showed that the PANI/WO_3_ was reusable and that it could also maintain relatively high activity after several experiments. The negligible loss of activity could be observed after each test, and 65% MB degradation was still achieved after four repeats. The decrease of the photocatalytic activity after each experiment was probably due to the inactivation of the polyaniline surface. Two ways to reactivate a photocatalyst were performed. At the beginning, the photocatalyst was kept at 0.9 V for 5 min in order to remove the photodegradation products from the surface of the photocatalyst. Next, the recovered photocatalyst was utilized to the degradation process of MB again. The enhancement of the photodecomposition rate was not observed suggesting that the polyaniline should be reduced before another photocatalytic test. Thus, the FTO/PANI/WO_3_ electrode material was polarized at −0.4 V for 5 min followed by the photocatalytic test. It turned out that the photoactivity of the PANI/WO_3_ was recovered, suggesting that the organic part was responsible for the decrease of the material photoactivity.

The possible mechanism of the photocatalytic performance of PANI/WO_3_ composite film was summed up and presented in a form of a scheme, see Figure 11.Under simulated solar light irradiation, both parts of the composite can be photoexcited. PANI absorbs photons to induce HOMO–LUMO transition, transporting the excited-state electrons from the LUMO-orbital to WO_3_ (LUMO of p-type PANI is more negative than the CB of inorganic n-type exfoliated WO_3_). The energy band alignment enables the transfer of the photogenerated electrons. According to previous reports about PANI photocatalytic properties, holes rather do not participate in the direct oxidation of organic pollutants adsorbed on the photocatalyst, however, the photoexcitation may lead to the formation of hydroxyl radicals [48]. In the case of WO_3_, holes from the valence band are mostly consumed for direct oxidation of the adsorbed dye. However, their role is more complex. One can expect the formation of hydroxyl radicals on WO_3_, as well. On the other hand, holes can migrate to HOMO of polyaniline or simply oxidize it. Since the presence of benzoquinone during the photocatalytic degradation of the dye is almost negligible, photoexcited electrons do not participate efficiently in the formation of superoxides. Thus, the electrons can have the opposite effect to the holes, i.e., take part in the recombination with the holes of the HOMO PANI or reduce the polymeric part of the film. Probably, there is a dynamic equilibrium that shifts over time to the oxidized form of polyaniline. This phenomenon negatively affects the degradation efficiency of MB in the subsequent degradation cycles. Applying a cathodic potential to the electrode reverses this effect, regenerating the photocatalyst, while anodic polarization decreases the photocatalyst performance. Consequently, WO_3_ and PANI work synergistically for the generation of AOS (advanced oxidation species) by hampering the electron-hole recombination and boosting charge separation, which led to the formation of an effective photocatalyst for the enhanced degradation of MB.

## 4. Conclusions

This report is devoted to the design and investigation of a novel composite electrode material, consisting of PANI and an exfoliated WO_3_. PANI/WO_3_ was electrochemically deposited on FTO glass from a solution containing aniline and an exfoliated WO_3_. Electrochemical tests of the PANI/WO_3_ indicated that it may act as an electrode material for energy storage application, and the results were more satisfactory in comparison with PANI/Cl electrode material. Charging/discharging performance in a three-electrode configuration evidenced that the material was stable in over 10,000 cycles, giving areal capacitance of 215 mF cm^−2^ with capacity retention of 74%. Moreover, most of the charge was stored by a capacitive mechanism. The fraction of a capacitive contribution of the surface processes was over 70% and was mainly affected by the presence of an exfoliated WO_3_. The practical utilization of PANI/WO_3_ electrode material was determined by a construction of a symmetric supercapacitor. Such device after 10,000 cycles exhibited high capacitance retention of 69% and 12.25 Wh kg^−1^ and 1075 W kg^−1^ energy and power density, respectively.

Furthermore, PANI/WO_3_ electrode material served as a catalyst in a photodegradation of organic compounds and was able to degrade dyes with the efficiency of over 84%. The investigation confirmed that the addition of tungsten oxide played a crucial role in the degradation process due to the utilization of holes, originating from the valence band of the inorganic WO_3_. As a final remark one may conclude that PANI/WO_3_ electrode material exhibited broad spectrum of applications from energy storage devices to photodegradation of organic compounds.

## Figures and Tables

**Figure 1 materials-13-05781-f001:**
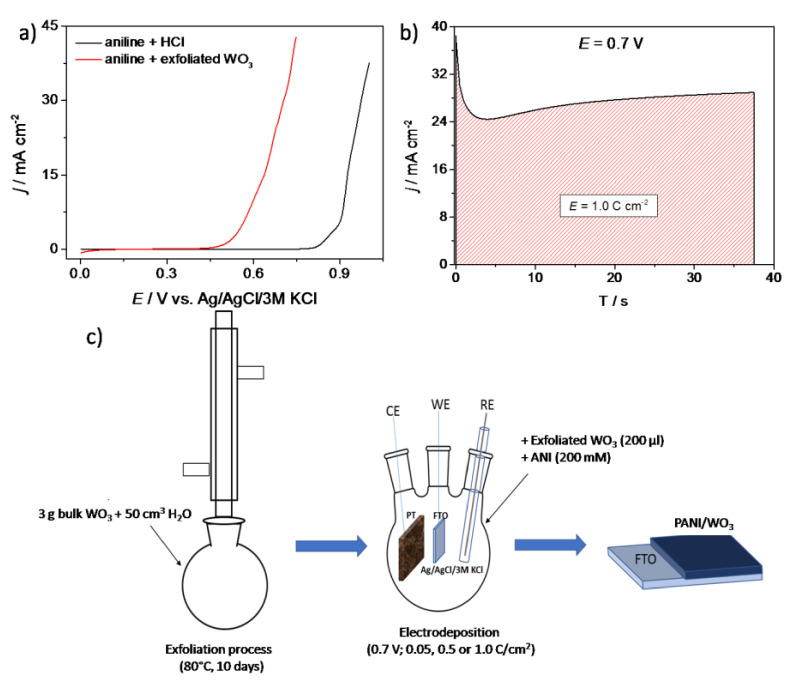
(**a**) The linear sweep voltammetry curves obtained during polarization of FTO electrodes in an aqueous electrolyte containing aniline/HCl (black curve) and aniline/exfoliated WO_3_ (red curve); (**b**) a chronoamperometry curve recorded during PANI/WO_3_electrodeposition and (**c**) the diagram depicting the preparation process of the FTO/PANI/WO_3_ composite.

**Figure 2 materials-13-05781-f002:**
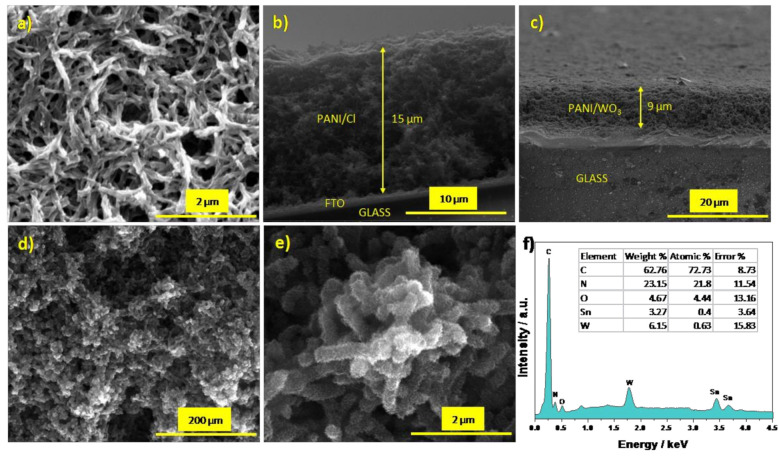
(**a**) SEM image of FTO/PANI/Cl; a cross-section SEM image of (**b**) FTO/PANI/Cl and (**c**) FTO/PANI/WO_3_; (**d**,**e**) SEM images of FTO/PANI/WO_3_ and (**f**) EDX spectrum of FTO/PANI/WO_3_ (inset: a table with listed elements content).

**Figure 3 materials-13-05781-f003:**
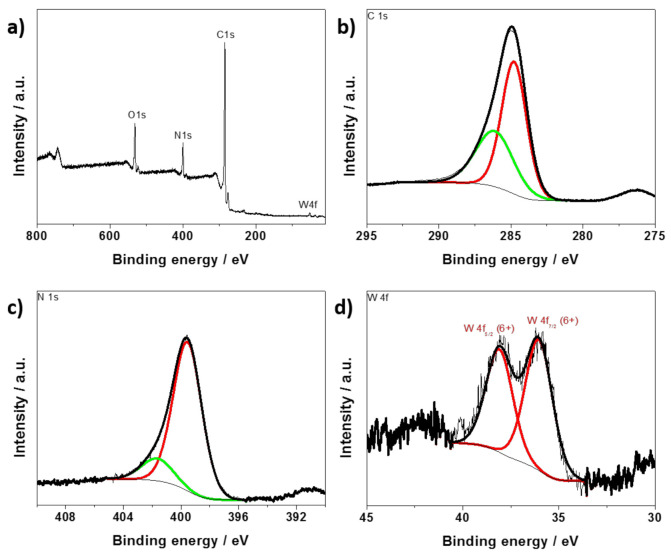
(**a**) XPS wide-scan spectrum of FTO/PANI/WO_3_; high-resolution XPS spectra recorded for (**b**) C1s, (**c**) N1s and (**d**) W4f region.

**Figure 4 materials-13-05781-f004:**
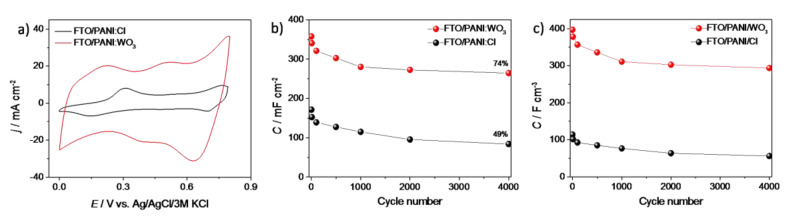
(**a**) The cyclic voltammetry curves of FTO/PANI/Cl and FTO/PANI/WO_3_ in 1 M H_2_SO_4_, scan rate 50 mV s^−1^; (**b**) the areal capacitance and (**c**) the volumetric capacitance vs. number of cycle plot for FTO/PANI/Cl and FTO/PANI/WO_3_ electrode materials (charge/discharge current density 2 mA cm^−2^, polarized from 0 to 0.8 V vs. Ag/AgCl/3 M KCl).

**Figure 5 materials-13-05781-f005:**
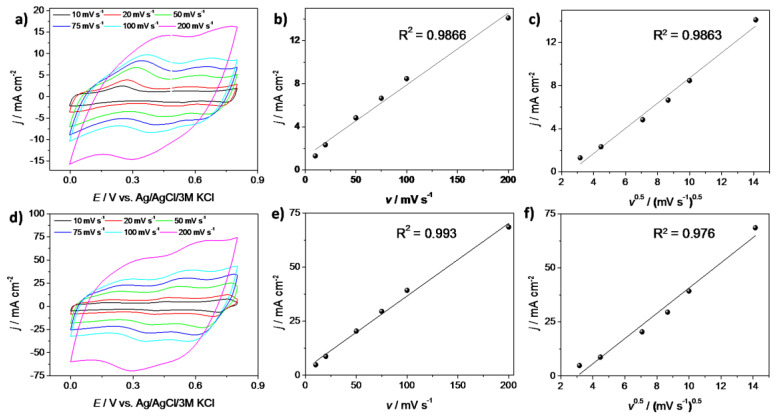
Cyclic voltammetry (CV) curves of (**a**) FTO/PANI/Cl and (**d**) FTO/PANI/WO_3_ electrodes in 1 M H_2_SO_4_ at different scan rates; (**b**,**e**) plots of j = *f*(v) and (**c**,**f**) j = *f*(v^1/2^) at E = 0.6 V for FTO/PANI/Cl and FTO/PANI/WO_3_, respectively.

**Figure 6 materials-13-05781-f006:**
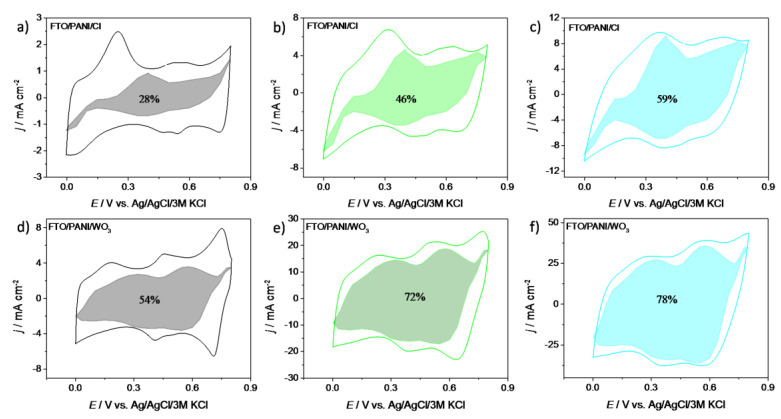
Charge storage contributions at (**a**) 10 mV s^−1^, (**b**) 50 mV s^−1^ and (**c**) 100 mV s^−1^ of FTO/PANI/Cl electrode material and at (**d**) 10 mV s^−1^, (**e**) 50 mV s^−1^ and (**f**) 100 mV s^−1^ of FTO/PANI/WO_3_ electrode material.

**Figure 7 materials-13-05781-f007:**
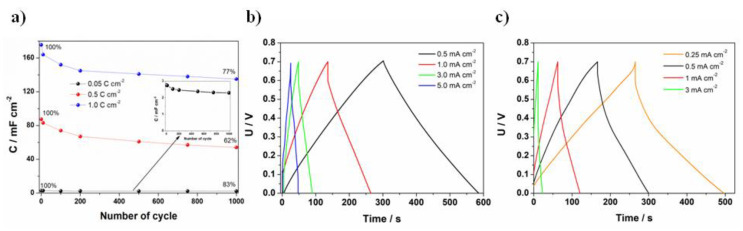
(**a**)The capacitance vs. number of the cycle plot recorded for a symmetric supercapacitor FTO/PANI/WO_3_; 0.05, 0.5 and 1 C cm^−2^ stands for charge consumed during electrodeposition of one electrode material; charge and discharge curves at different current densities applied for (**b**) PANI/WO_3_ symmetric supercapacitor and (**c**) PANI/Cl symmetric supercapacitor.

**Figure 8 materials-13-05781-f008:**
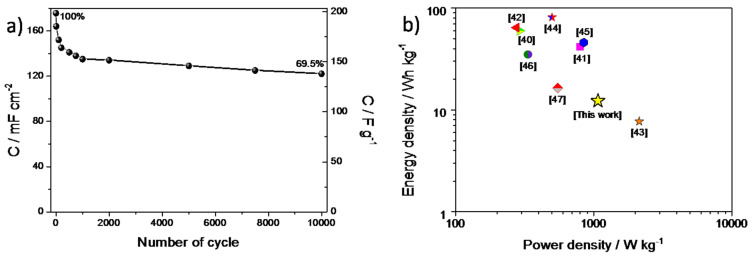
**(a**) The capacitance (areal and gravimetric) vs. number of cycle plot recorded for a PANI/WO_3_-based symmetric supercapacitor and (**b**) Ragone plot for a FTO/PANI/WO_3_ symmetric supercapacitor.

**Figure 9 materials-13-05781-f009:**
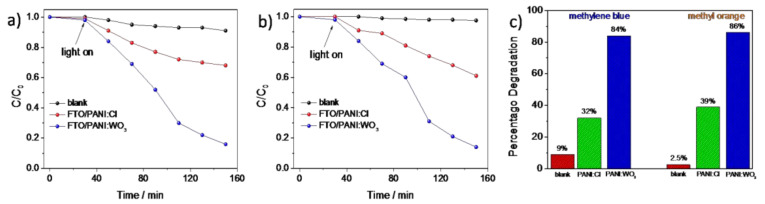
Comparison of the photocatalytic degradation of (**a**) MB and (**b**) MO in the presence ofPANI/Cl and PANI/WO_3_ composite photocatalysts under sunlight irradiation and (**c**) diagram showing the results of photodegradation.

**Figure 10 materials-13-05781-f010:**
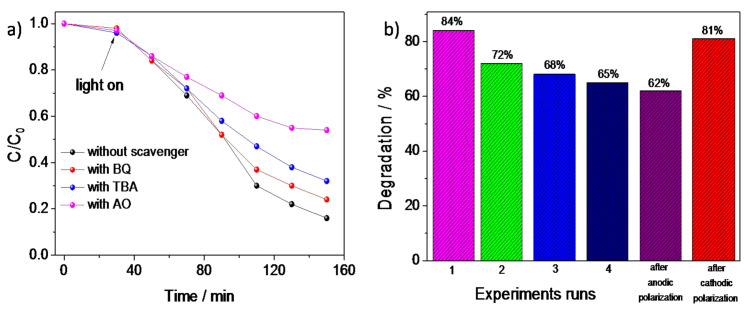
**(a**) The influence of different scavengers on photodegradation efficiency of MB and (**b**) the degradation efficiency of MB in the function of experimental repeats.

**Figure 11 materials-13-05781-f011:**
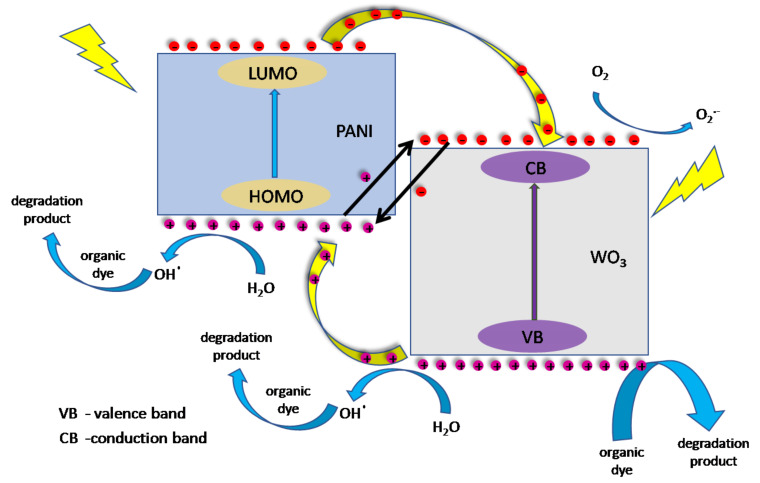
Schematic mechanism of a dye photodecomposition by the FTO/PANI/WO_3_ photocatalyst.

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
