# Peer review of "An Aqueous Exfoliation of WO3 as a Route for Counterions Fabrication—Improved Photocatalytic and Capacitive Properties of Polyaniline/WO3Composite"

_materials, 2020, doi:10.3390/ma13245781_

Round 1
Reviewer 1 Report
In this work, a PANI/WO3 composite film was electrochemically polymerized using the exfoliated WO3 as counterions. SEM and XPS were used to confirm the successful hybridization of PANI and WO3. Due to the synergistic effect of PANI and WO3, improved electrochemical and photocatalytic performances were achieved in the composite films. Before acceptance, the authors should answer the following questions.
1) The mass loading of PANI/Cl and PANI/WO3 electrodes should be provided and normalized. Though both films were obtained using the same polymerization charge, the mass loading of the materials could be different. Therefore, the authors should exhibit the gravimetric capacitance of PANI/Cl and PANI/WO3 electrodes based on a similar mass loading.
2) The authors should discuss the function of WO3 in the composite in more detail.
3) The charge/discharge profiles and the rate performance of PANI/Cl and PANI/WO3 electrodes should be given.
4) If WO3 was doped in PANI as the counterion, the interaction between PANI and WO3 should be realized. However, XPS indicated that the valence state of W in WO3 is 6+, that is we did not observe the interaction between PANI and the incorporated WO3. Except for the XPS data, could the authors give other proof to illustrate the successful doping of WO3 in PANI?
Reviewer 2 Report
The paper reports a hybrid structure of PANI and WO3-x as a supercapacitor and photocatalyst. The obtained results indicate that the materials are promising for stable supercapacitors. The paper can be interesting for readers. I recommend the paper for publication after addressing the following comments.
- The full names for some abbreviations such as fluorine-doped tin oxide (FTO), aniline (ANI), ethylenedioxythiophene (EDOT) should be given for the first time of their appearance in the main text. The name of the electrode given in the caption of Figure 7 should be revised to be consistent with the writing of sample names in the other parts of the paper (FTO/PANI:Clà FTO/PANI/Cl, FTO/PANI:WO3 à FTO/PANI/WO3).
- The capacitance was measured for both 3 and 2-electrode configurations and the areal capacitance was calculated, the results indicate a higher value for FTO/PANI/WO3 than FTO/PANI/Cl. The surface area and the thickness of the active layer of the two electrodes can be different. Therefore, please provide the cross-sectional image of FTO/PANI/Cl and the volumetric capacitance for the two electrodes.
- Quantitative weight composition of the obtained PANI/WO3 after deposition should be given. This is because the electrode performance can depend on the electrode composition.
- Can the authors comment on the better capacitance retention obtained when using 2-electrode configurations compared with 3-electrode configuration?
- The degradation of the capacitance for both electrode configurations occurred most significantly in the first 1000 cycles. Did the authors observe the change in the morphology or leaching of WO3 or decomposition of PANI after 1000 cycles?
- Adding an illustration of the photocatalyst dye degradation by PANI/WO3 is recommended to support the discussion of the hole generation from VB of WO3.
Reviewer 3 Report
see the attached file.

Author Response
Thank you for your kind opinion
Reviewer 4 Report
In this interesting article Nowak and co-workers study to what extent electrodeposited PANI/WO3 layers can be used as photoactive material for the decomposition of methylene blue and methyl orange.
The paper is well written, the research well and diligently carried out, and I can only suggest minor improvements as follow:
1) do the authors have an explanation as to why the ANI oxidation conditions become less harsh (lower biases) on exfoliated WO3? Has it to do with kinetics or thermodynamics? Further (perhaps speculative but it may seed ideas for further research), is it possible that the oxidation of OH- to OH* (hydroxyde to hydroxyls) is somewhat involved? This would explain the lower oxidation potential for the oxidation. At the interface between hydrophobic materials and water there are strong electric fields and very large molarities of unbalanced OH-. There is a very recent nat. commun. on facilitated oxidation reactions at water-hydrophobic interfaces
2) Following on the point above, would it be possible to test for the presence of OH* radicals? Maybe by fluorescence with some OH* sensitive dye?
3) Superoxide scavengers. Is it expected to form superoxide by oxygen reduction in water? I think no, in water the oxygen reduction become a two electron process. It was not clear to me the discussion on superoxyde in the bit relative to the degradation of MB. Overall an excellent and interesting work
Round 2
Reviewer 1 Report
The authors have answered all of my questions. I suggest the acceptance of this manuscript.